# Prediction of the Physical Properties of a Structural Member by the Impact Hammer Test

**DOI:** 10.3390/s22186762

**Published:** 2022-09-07

**Authors:** Eun-Taik Lee, Yu-Sik Hong, Hee-Chang Eun

**Affiliations:** 1Department of Architectural Engineering, Chung-Ang University, Seoul 06974, Korea; 2Department of Architectural Engineering, Kangwon National University, Chuncheon 24341, Korea

**Keywords:** impact hammer test, frequency response function, accelerometer, elastic modulus, resonance frequency, measurement

## Abstract

The frequency response function (FRF) in the frequency domain is a black box used to collect physical information and to indicate the modal characteristics of a dynamic system. Analyzing the collected FRF data through the impact hammer test, dynamic parameters, such as stiffness, mass, and the damping matrix, can be estimated. By extracting and analyzing the FRFs within certain ranges of the lowest few resonance frequencies, this study presents a nondestructive method to estimate the dynamic parameters and the material properties. Updating of the dynamic parameters and material properties is a crucial process for the subsequent design and maintenance. This study presents a method to estimate the physical properties of structural members using measured FRF data and generalized inverse. By extracting and analyzing the FRFs within certain ranges of the lowest few resonance frequencies, the dynamic parameters were predicted. It was observed in numerical experiments that the proposed method could properly estimate the elastic modulus and dynamic parameters of steel beams, although the results were affected by the extracted FRF ranges. The physical properties were close to more accurate values in taking the FRFs at more resonance frequencies, as the member was flexible. The proposed method was also extended to a nondestructive test for an estimation of the compressive strength of concrete. However, it faced difficulty due to the external noise contained in the measured data. It was found sin the nondestructive test that the proposed technique was affected by external noise, unlike a simple steel beam. The concrete strength could be predicted by taking the FRFs in a wide frequency range containing the lowest two resonance frequencies and by averaging the repeated test results.

## 1. Introduction

A structure deteriorates depending on the period of use, the surrounding environment, and the management condition. The structural service life should be improved through repair and reinforcement. The deterioration of the structure leads to a change in dynamic parameters such as mass, stiffness, and damping, as well as modal parameters such as frequency and mode shape. The physical properties of concrete, such as the elastic modulus and compressive strength, are predicted from the physical parameters. The decision regarding the implementation of reinforcement is made by evaluating the structural performance through appropriate methods. Structural system identification techniques have been used in the process of reducing the deviation between structural systems and their design models [1,2], and they have been applied in structural health monitoring to evaluate the structural performance [3,4,5]. The estimation techniques of dynamic parameters are important in system identification and structural health monitoring. Most of the analysis techniques estimate the dynamic modal characteristics [6,7,8,9].

Peng et al. [10] observed the efficiency of constrained observability techniques for structural system identification using dynamic data. Chen et al. [11] proposed a diagnosis method for incipient material grade using a sequential probability ratio test and a nonlinear ultrasonic method. Chen and Li [12] presented an identification method to identify the microcracks in a mechanical system using heterolateral colinear mixed-frequency ultrasound. Bai et al. [13] developed a damage simulation model to predict the fatigue resistance using spring hinges.

It is important to determine the physical parameters through measurement for the subsequent analysis and maintenance of structures. The parameter matrices should be updated using the measured data and initial parameters during the design, and the updated parameters can be utilized as the fundamental information of the existing structures [14,15,16,17].

With the advent of sensitive measurement sensors, several methods for estimating the physical properties of structural members and materials have been proposed. They can be estimated by analyzing vibration data through dynamic tests rather than static tests. As one of these methods, the impact hammer test can estimate the dynamic parameters from measured frequency response function (FRF) data [18,19,20,21]. Impact hammer testing or modal testing is an experimental method to extract dynamic information using the impact hammer and an accelerometer. Techniques in the frequency domain for predicting the dynamic characteristics can be used to evaluate the transfer function of the input and output relationship according to the excitation frequency.

Matsubara et al. [22] estimated the mode parameters using the ratio of the real and imaginary parts of the FRF data. Lee and Richardson [23] used a parameter curve fitting method to estimate the unknown mode parameters from FRF measurements. Assuming that the modal parameters were random variables, Doebling and Farrar [24] predicted the modal parameters based on the random errors associated with averaged FRF estimates. Shih et al. [25] introduced a complex mode indication function from the FRF matrix and proposed a method for estimating parameters in the spatial domain. De Rozario and Oomen [26] developed a variable linear parametric FRF model from experiments. Machado et al. [27] proposed a mode update algorithm to predict material and geometric parameters such as Poisson’s ratio and elastic modulus. Lee and Kim [28] identified the damping characteristics of a system by inverting the FRF matrix to the dynamic stiffness matrix and comparing their real and imaginary parts with parameter matrices. Fritzen [29] proposed a method to calculate the parameter matrices from measured input and output data based on the instrumental variable method. Rahmatalla et al. [30] evaluated a method of estimating parameter values from the amount of change in FRF data using a generalized inverse matrix. Tam et al. [31] found the elastic properties by updating the predicted FRFs with reference to the experimental FRFs depending on different boundary conditions.

One of the tests to evaluate the physical properties of existing structures is a nondestructive test to predict concrete’s strength. The deterioration of reinforced concrete (RC) structures should be evaluated by field investigations such as destructive and nondestructive tests (NDTs). The compressive strength of concrete has been estimated by performing NDTs such as the rebound hammer test and the ultrasonic pulse velocity test. The rebound hammer test estimates the concrete’s strength by measuring the rebounding coefficient when hitting the concrete member surface. The ultrasonic pulse velocity test measures the time required for a sound wave to pass between the sender and the receiver and predicts the concrete’s strength via the magnitude of the sound speed. The rebound hammer test is affected by various factors: smoothness of the surface, size, and shape of the specimen; moisture condition of the concrete, etc. The ultrasonic pulse velocity test is much more limited, owing to the large number of related parameters between strength and pulse velocity [32].

This study considered a method for estimating the physical properties based on the vibration response of the structure by external excitation. The physical parameters, such as stiffness, mass, and damping, and the physical properties, namely, the elastic modulus and compressive strength, of concrete were estimated using the measured FRFs. The FRF data were measured by the impact hammer test, and the parameters were derived using a generalized inverse matrix, taking the square root of the sum of the squares (SRSS) of the parameters at each resonance frequency. The dynamic parameter matrices were predicted by extracting the FRF data in the neighborhood of the lowest several resonance frequencies, and the accuracy was evaluated. The validity and utilization of the proposed method were examined in numerical examples and a nondestructive test. It was observed that the accuracy in predicting the physical properties was sensitive to the FRFs within the selected excitation frequency ranges and the flexibility degree of the structural member. The proposed method had difficulty in estimating the compressive strength of concrete owing to the measured data being contaminated by external noise. The concrete strength could be predicted by taking the FRFs in a wide frequency range containing the lowest two resonance frequencies and averaging the repeated test results.

In Section 2, the formulation for estimating the dynamic parameters and material properties of the structural members is derived. Section 3 exhibits two numerical experiments and an indoor test to illustrate the proposed method. Finally, Section 4 summarizes the analytical and numerical results of this study.

## 2. Formulation

A structural member vibrates due to the fact of external excitation, and the vibration response depends on the dynamic characteristics. Thus, the dynamic parameters are inversely predicted by analyzing the dynamic responses in the frequency domain rather than in the time domain. The transfer function, which is the relationship between the input and output, is the black box used to explain the dynamic characteristics. The resonance frequency and the corresponding vibration mode are estimated from the FRF transfer function [33].

The equation of motion for a finite element model with *n* degrees of freedom in the time domain is expressed as:(1)$$M\ddot{u}+C\dot{u}+Ku=p\left(t\right)$$
where **M**, **C**, and **K** represent the n×n mass, damping, and stiffness matrices, respectively. **u** represents the displacement response, the dot on the displacement represents the time derivative, and **p** represents the disturbance of a dynamic load. Substituting $u=Ue^{i\Omega t}$ and $p=Pe^{i\Omega t}$ in Equation (1), it is transformed in the equation of motion in the frequency domain as:(2)$$\left[-\Omega^{2}M+i\Omega C+K\right]U\left(\Omega \right)=P\left(\Omega \right)$$
where Ω is the excitation frequency, and $U\left(\Omega \right)$ and $P\left(\Omega \right)$ are the displacement and force vectors in the frequency domain, respectively. $i=\sqrt{-1}\;$ and $\left[-\Omega^{2}M+i\Omega C+K\right]\;$ represent the dynamic stiffness matrix. If the displacement-based FRF $H_{d}\left(\Omega \right)$ is expressed in Equation (2) as ${\left[-\Omega^{2}M+i\Omega C+K\right]}^{-1}$, it can be written as follows:(3)$$U\left(\Omega \right)=H_{d}\left(\Omega \right)P\left(\Omega \right)$$
where $H_{d,ij}$ represents the FRF receptance to indicate the displacement response at position *i* due to the unit impulse at position *j*.

From the curve to represent the magnitude of the FRF receptance or inertance according to the excitation frequency, the resonance frequencies and the corresponding FRFs are obtained. The mode shape curves can be described by connecting the maximum magnitude of the FRFs corresponding to the same resonance frequencies at all nodes. Similarly, the dynamic parameter matrices can be estimated by collecting and analyzing the FRFs within certain ranges of the resonance frequencies.

Using the acceleration response A by the accelerometer instead of the displacement response U, the relationship $A=-\Omega^{2}U$ is inserted into Equation (3). This yields:(4)$$A\left(\Omega \right)=-\Omega^{2}H_{d}\left(\Omega \right)P\left(\Omega \right)=H_{A}\left(\Omega \right)P\left(\Omega \right)$$
where $H_{A,ij}$ denotes the FRF inertance to represent the acceleration response at position *i* due to the unit impulse at position *j* in the frequency domain.

As shown in Figure 1, fixing the position of the impact hammer at node *l*, installing the accelerometers at all *n* positions, and measuring FRFs, they are written by:(5)$$\begin{array}{c} H_{A}^{1l}\left(\Omega \right)=H_{A,re}^{1l}\left(\Omega \right)+iH_{A,im}^{1l}\left(\Omega \right)\\ H_{A}^{2l}\left(\Omega \right)=H_{A,re}^{2l}\left(\Omega \right)+iH_{A,im}^{2l}\left(\Omega \right)\\ \vdots \\ H_{A}^{\left(n-1\right)l}\left(\Omega \right)=H_{A,re}^{\left(n-1\right)l}\left(\Omega \right)+iH_{A,im}^{\left(n-1\right)l}\left(\Omega \right)\\ H_{A}^{nl}\left(\Omega \right)=H_{A,re}^{nl}\left(\Omega \right)+iH_{A,im}^{nl}\left(\Omega \right) \end{array}$$
where the subscripts *re* and *im* indicate the real part and the imaginary part of the complex number, respectively.$\;H_{A}^{ml}\left(\Omega \right)$ is the FRF inertance at node *m* due to the action of the impact hammer on node *l* at the disturbance frequency Ω.

The parameter matrices of a finite element model are symmetric and positive definite. However, the elements in the parameter matrices updated using the measured data rarely match one-to-one with those in the initial matrices of the finite element model. They are not symmetric matrices, because they are predicted by measured data and the external noise contained within them. Nevertheless, the parameter matrices should be corrected for the subsequent analysis, and they were derived in this work.

The result is closely related to the physical properties at node *l*, $M^{lm}$, $C^{lm}$, and $K^{lm}$ $\left(m=1,\;2,\ldots ,\;\left(n-2\right),\;\left(n-1\right),\;n\right)$, which represent the *l*th row vectors of the mass, damping, and stiffness matrices, respectively. Multiplying $\left[-\Omega^{2}M^{lm}+i\Omega C^{lm}+K^{lm}\right]\frac{1}{\Omega^{2}},\;\left(m=1,\;2,\ldots ,\;\left(n-2\right),\;\left(n-1\right),\;n\right)$, and separating the result into a real part and a complex part, we obtained the following expression:(6)$$\begin{array}{c} \left[\begin{array}{cc} -\Omega^{2}H_{A,re}^{1l}\left(\Omega \right) & \begin{array}{cc} -\Omega H_{A,im}^{1l}\left(\Omega \right) & H_{A,re}^{1l}\left(\Omega \right) \end{array}\\ -\Omega^{2}H_{A,im}^{1l}\left(\Omega \right) & \begin{array}{cc} \Omega H_{A,re}^{1l}\left(\Omega \right) & H_{A,im}^{1l}\left(\Omega \right) \end{array} \end{array}\right]\left[\begin{array}{c} M^{l1}\\ C^{l1}\\ K^{l1} \end{array}\right]=\left[\begin{array}{c} \Omega^{2}\\ 0 \end{array}\right]\\ \left[\begin{array}{cc} -\Omega^{2}H_{A,re}^{ll}\left(\Omega \right) & \begin{array}{cc} -\Omega H_{A,im}^{ll}\left(\Omega \right) & H_{A,re}^{ll}\left(\Omega \right) \end{array}\\ -\Omega^{2}H_{A,im}^{ll}\left(\Omega \right) & \begin{array}{cc} \Omega H_{A,re}^{ll}\left(\Omega \right) & H_{A,im}^{ll}\left(\Omega \right) \end{array} \end{array}\right]\left[\begin{array}{c} M^{ll}\\ C^{ll}\\ K^{ll} \end{array}\right]=\left[\begin{array}{c} \Omega^{2}\\ 0 \end{array}\right]\\ \vdots \\ \left[\begin{array}{cc} -\Omega^{2}H_{A,re}^{nl}\left(\Omega \right) & \begin{array}{cc} -\Omega H_{A,im}^{nl}\left(\Omega \right) & H_{A,re}^{nl}\left(\Omega \right) \end{array}\\ -\Omega^{2}H_{A,im}^{nl}\left(\Omega \right) & \begin{array}{cc} \Omega H_{A,re}^{nl}\left(\Omega \right) & H_{A,im}^{nl}\left(\Omega \right) \end{array} \end{array}\right]\left[\begin{array}{c} M^{ln}\\ C^{ln}\\ K^{ln} \end{array}\right]=\left[\begin{array}{c} \Omega^{2}\\ 0 \end{array}\right] \end{array}$$
where the superscript *ln* indicates the element in the *l*th row and *n*th column.

It was observed that the parameters were expressed as a function of the external excitation frequency. Extracting the FRF data within an arbitrary range of the lowest *r* resonance frequencies, they were expanded as follows:

(1) *s* sets of FRFs within the first resonance frequency ($\Omega_{1}$):
(7a)$$\begin{array}{ccc} {\left[\begin{array}{ccc} \begin{array}{c} -\Omega_{11}^{2}H_{A,re}^{jl}\left(\Omega_{11}\right)\\ -\Omega_{11}^{2}H_{A,im}^{jl}\left(\Omega_{11}\right)\\ -\Omega_{12}^{2}H_{A,re}^{jl}\left(\Omega_{12}\right) \end{array} & \begin{array}{c} -\Omega_{11}H_{A,im}^{jl}\left(\Omega_{11}\right)\\ \Omega_{11}H_{A,re}^{jl}\left(\Omega_{11}\right)\\ -\Omega_{12}H_{A,im}^{jl}\left(\Omega_{12}\right) \end{array} & \begin{array}{c} H_{A,re}^{jl}\left(\Omega_{11}\right)\\ H_{A,im}^{jl}\left(\Omega_{11}\right)\\ H_{A,re}^{jl}\left(\Omega_{12}\right) \end{array}\\ \begin{array}{c} -\Omega_{12}^{2}H_{A,im}^{jl}\left(\Omega_{12}\right)\\ \vdots \\ -\Omega_{1s}^{2}H_{A,re}^{jl}\left(\Omega_{1s}\right) \end{array} & \begin{array}{c} \Omega_{12}H_{A,re}^{jl}\left(\Omega_{12}\right)\\ \vdots \\ -\Omega_{1s}H_{A,im}^{jl}\left(\Omega_{1s}\right) \end{array} & \begin{array}{c} H_{A,im}^{jl}\left(\Omega_{12}\right)\\ \vdots \\ H_{A,re}^{jl}\left(\Omega_{1s}\right) \end{array}\\ -\Omega_{1s}^{2}H_{A,im}^{jl}\left(\Omega_{1s}\right) & \Omega_{1s}H_{A,re}^{jl}\left(\Omega_{1s}\right) & H_{A,im}^{jl}\left(\Omega_{1s}\right) \end{array}\right]}_{} & {\left[\begin{array}{c} M^{lj}\\ C^{lj}\\ K^{lj} \end{array}\right]}^{\Omega_{1}}={\left[\begin{array}{c} \begin{array}{c} \Omega_{11}^{2}I\\ 0\\ \Omega_{12}^{2}I \end{array}\\ \begin{array}{c} 0\\ \vdots \\ \Omega_{1s}^{2}I \end{array}\\ 0 \end{array}\right]}_{} & ,\\ j=1,2,\ldots ,\left(n-1\right),n \end{array}$$

(2) *p* sets of FRFs within the *r*th resonance frequency ($\Omega_{r}$):(7b)$$\begin{array}{ccc} {\left[\begin{array}{ccc} \begin{array}{c} -\Omega_{r1}^{2}H_{A,re}^{jl}\left(\Omega_{r1}\right)\\ -\Omega_{r1}^{2}H_{A,im}^{jl}\left(\Omega_{r1}\right)\\ -\Omega_{r2}^{2}H_{A,re}^{jl}\left(\Omega_{r2}\right) \end{array} & \begin{array}{c} -\Omega_{r1}H_{A,im}^{jl}\left(\Omega_{r1}\right)\\ \Omega_{r1}H_{A,re}^{jl}\left(\Omega_{r1}\right)\\ -\Omega_{r2}H_{A,im}^{jl}\left(\Omega_{r2}\right) \end{array} & \begin{array}{c} H_{A,re}^{jl}\left(\Omega_{r1}\right)\\ H_{A,im}^{jl}\left(\Omega_{r1}\right)\\ H_{A,re}^{jl}\left(\Omega_{r2}\right) \end{array}\\ \begin{array}{c} -\Omega_{r2}^{2}H_{A,im}^{jl}\left(\Omega_{r2}\right)\\ \vdots \\ -\Omega_{rp}^{2}H_{A,re}^{jl}\left(\Omega_{rp}\right) \end{array} & \begin{array}{c} \Omega_{r2}H_{A,re}^{jl}\left(\Omega_{r2}\right)\\ \vdots \\ -\Omega_{r(mr)}H_{A,im}^{jl}\left(\Omega_{rp}\right) \end{array} & \begin{array}{c} H_{A,im}^{jl}\left(\Omega_{r2}\right)\\ \vdots \\ H_{A,re}^{jl}\left(\Omega_{rp}\right) \end{array}\\ -\Omega_{rp}^{2}H_{A,im}^{jl}\left(\Omega_{rp}\right) & \Omega_{rp}H_{A,re}^{jl}\left(\Omega_{rp}\right) & H_{A,im}^{jl}\left(\Omega_{rp}\right) \end{array}\right]}_{} & {\left[\begin{array}{c} M^{lj}\\ C^{lj}\\ K^{lj} \end{array}\right]}^{\Omega_{r}}={\left[\begin{array}{c} \begin{array}{c} \Omega_{r1}^{2}I\\ 0\\ \Omega_{r2}^{2}I \end{array}\\ \begin{array}{c} 0\\ \vdots \\ \Omega_{rp}^{2}I \end{array}\\ 0 \end{array}\right]}_{} & ,\\ j=1,2,\ldots ,\left(n-1\right),n \end{array}$$
where the subscripts 1 and *s* in $\Omega_{1s}$ denote the 1st resonance frequency and the corresponding *s* sets of FRF data, respectively. $\Omega_{rp}$ is the *r*th resonance frequency and the *p* sets of FRF data. Equation (7a,b) represent simultaneous equations with respect to the dynamic parameters at each resonance frequency.

Utilizing the generalized inverse and solving it with respect to the physical parameters $M^{lj}$, $C^{lj}$, and $K^{lj}$ yields:(8a)$$\begin{array}{cccc} {\left[\begin{array}{c} M^{lj}\\ C^{lj}\\ K^{lj} \end{array}\right]}^{\Omega_{1}}= & {\left[\begin{array}{ccc} \begin{array}{c} -\Omega_{11}^{2}H_{A,re}^{jl}\left(\Omega_{11}\right)\\ -\Omega_{11}^{2}H_{A,im}^{jl}\left(\Omega_{11}\right)\\ -\Omega_{12}^{2}H_{A,re}^{jl}\left(\Omega_{11}\right) \end{array} & \begin{array}{c} -\Omega_{11}H_{A,im}^{jl}\left(\Omega_{11}\right)\\ \Omega_{11}H_{A,re}^{jl}\left(\Omega_{11}\right)\\ -\Omega_{12}H_{A,im}^{jl}\left(\Omega_{12}\right) \end{array} & \begin{array}{c} H_{A,re}^{jl}\left(\Omega_{11}\right)\\ H_{A,im}^{jl}\left(\Omega_{11}\right)\\ H_{A,re}^{jl}\left(\Omega_{12}\right) \end{array}\\ \begin{array}{c} -\Omega_{12}^{2}H_{A,im}^{jl}\left(\Omega_{12}\right)\\ \vdots \\ -\Omega_{1(m1)}^{2}H_{A,re}^{jl}\left(\Omega_{1s}\right) \end{array} & \begin{array}{c} \Omega_{12}H_{A,re}^{jl}\left(\Omega_{12}\right)\\ \vdots \\ -\Omega_{1m}H_{A,im}^{jl}\left(\Omega_{1s}\right) \end{array} & \begin{array}{c} H_{A,im}^{jl}\left(\Omega_{12}\right)\\ \vdots \\ H_{A,re}^{jl}\left(\Omega_{1s}\right) \end{array}\\ -\Omega_{1(m1)}^{2}H_{A,im}^{jl}\left(\Omega_{1s}\right) & \Omega_{1m}H_{A,re}^{jl}\left(\Omega_{1s}\right) & H_{A,im}^{jl}\left(\Omega_{1s}\right) \end{array}\right]}^{+} & {\left[\begin{array}{c} \begin{array}{c} \Omega_{11}^{2}I\\ 0\\ \Omega_{12}^{2}I \end{array}\\ \begin{array}{c} 0\\ \vdots \\ \Omega_{1s}^{2}I \end{array}\\ 0 \end{array}\right]}_{} & ,\\ & j=1,2,\ldots ,\left(n-1\right),n\\ & \vdots \end{array}$$
(8b)$$\begin{array}{cccc} {\left[\begin{array}{c} M^{lj}\\ C^{lj}\\ K^{lj} \end{array}\right]}^{\Omega_{r}}= & {\left[\begin{array}{ccc} \begin{array}{c} -\Omega_{r1}^{2}H_{A,re}^{jl}\left(\Omega_{r1}\right)\\ -\Omega_{r1}^{2}H_{A,im}^{jl}\left(\Omega_{r1}\right)\\ -\Omega_{r2}^{2}H_{A,re}^{jl}\left(\Omega_{r2}\right) \end{array} & \begin{array}{c} -\Omega_{r1}H_{A,im}^{jl}\left(\Omega_{r1}\right)\\ \Omega_{r1}H_{A,re}^{jl}\left(\Omega_{r1}\right)\\ -\Omega_{r2}H_{A,im}^{jl}\left(\Omega_{r2}\right) \end{array} & \begin{array}{c} H_{A,re}^{jl}\left(\Omega_{r1}\right)\\ H_{A,im}^{jl}\left(\Omega_{r1}\right)\\ H_{A,re}^{jl}\left(\Omega_{r2}\right) \end{array}\\ \begin{array}{c} -\Omega_{r2}^{2}H_{A,im}^{jl}\left(\Omega_{r2}\right)\\ \vdots \\ -\Omega_{1(mr)}^{2}H_{A,re}^{jl}\left(\Omega_{rp}\right) \end{array} & \begin{array}{c} \Omega_{r2}H_{A,re}^{jl}\left(\Omega_{r2}\right)\\ \vdots \\ -\Omega_{r(mr)}H_{A,im}^{jl}\left(\Omega_{rp}\right) \end{array} & \begin{array}{c} H_{A,im}^{jl}\left(\Omega_{r2}\right)\\ \vdots \\ H_{A,re}^{jl}\left(\Omega_{rp}\right) \end{array}\\ -\Omega_{r(mr)}^{2}H_{A,im}^{jl}\left(\Omega_{rp}\right) & \Omega_{r(mr)}H_{A,re}^{jl}\left(\Omega_{rp}\right) & H_{A,im}^{jl}\left(\Omega_{rp}\right) \end{array}\right]}^{+} & {\left[\begin{array}{c} \begin{array}{c} \Omega_{r1}^{2}I\\ 0\\ \Omega_{r2}^{2}I \end{array}\\ \begin{array}{c} 0\\ \vdots \\ \Omega_{rp}^{2}I \end{array}\\ 0 \end{array}\right]}_{} & ,\\ & \left(j=1,2,\ldots ,\left(n-1\right)\right),n \end{array}$$
where “+” denotes the generalized inverse. Using the FRFs of the lowest *r* resonance frequencies and taking the SRSS of the parameters at each resonance frequency, the dynamic parameters of the *l*th row can be estimated by Equation (8a,b) as follows:(9)$$\begin{array}{c} M^{lj}=\sqrt{{\left[{\left(M^{lj}\right)}^{\Omega_{1}}\right]}^{2}+{\left[{\left(M^{lj}\right)}^{\Omega_{2}}\right]}^{2}+\ldots +{\left[{\left(M^{lj}\right)}^{\Omega_{r}}\right]}^{2}}\\ C^{lj}=\sqrt{{\left[{\left(C^{lj}\right)}^{\Omega_{1}}\right]}^{2}+{\left[{\left(C^{lj}\right)}^{\Omega_{2}}\right]}^{2}+\ldots +{\left[{\left(C^{lj}\right)}^{\Omega_{r}}\right]}^{2}}\\ K^{lj}=\sqrt{{\left[{\left(K^{lj}\right)}^{\Omega_{1}}\right]}^{2}+{\left[{\left(K^{lj}\right)}^{\Omega_{2}}\right]}^{2}+\ldots +{\left[{\left(K^{lj}\right)}^{\Omega_{r}}\right]}^{2}}\\ \left(j=1,2,\ldots ,\left(n-1\right)\right),n \end{array}$$

Moving the impact position to the other locations and repeating the same measurement, the parameters at the other rows can be estimated, and a full set of parameter matrices can be obtained.

The dynamic parameter matrices are estimated by Equations (8a,b) and (9). It can be seen that the accuracy of the parameters depends on the extracted FRFs within the selected frequency ranges. The selected number of the lowest resonance frequencies depends on the flexibility degree of the member. More flexible members require FRF data at more resonance frequencies. The dynamic parameters of stiff members can be predicted by the FRF data within a smaller number of resonance frequencies compared to flexible members.

The stiffness, $K^{lj}$, can be found via the elastic deflection curve in the *Textbook of Mechanics of Materials*. The stiffnesses of the both-end-fixed beam, $K_{bf}$, subjected to a concentrated force at the mid-span, as shown in Figure 2a, and the cantilevered beam,$\;K_{can}$, subjected to a concentrated force at the free end, as shown in Figure 2b, can be written, respectively, as:(10a)$$K_{bf}=\frac{24EIL}{x^{2}\left(\frac{3}{2}L^{2}-\frac{3}{2}Lx-\frac{L}{2}x\right)}$$
(10b)$$K_{can}=\frac{6EI}{x^{2}\left(3L-x\right)}$$
where E and I represent the elastic modulus and the moment of inertia, respectively. Moreover, L is the beam length, and x denotes the distance from the left end.

The cantilevered beam is more flexible than the both-end-fixed beam. Using the estimated stiffness parameters and Equation (10a,b), the elastic modulus at an arbitrary position can be explicitly determined. The stiffness matrix can be predicted by synthesizing the parameters at the selected resonance frequencies of the finite element model. At this time, the flexible member needs data at more resonance frequencies compared to the stiff member. Moreover, when the used material is concrete, the compressive strength of concrete, $f_{c}^{\prime }$, is estimated using the following equation, which expresses the relationship between the elastic modulus, $E_{c}$, and concrete’s strength, as determined in ACI 318-8:(11)$$\;\;f_{c}^{\prime }=\frac{E_{c}^{2}}{4700^{2}}\;\left(\mathrm{MPa}\right)$$

The validity of the proposed method, according to a nondestructive test to estimate the compressive strength of concrete, is evaluated in the following examples.

## 3. Examples

### 3.1. Both-End-Fixed Steel Beam

Numerical experiments were conducted to verify the validity of the proposed method. As shown in Figure 3, a 1 m long fixed-end beam with a cross-section of b×h=75×9 mm, an elastic modulus of $1.95\times 10^{5}\;\mathrm{MPa}$, and a weight per unit length of 7.86 kg/mm was utilized. A finite element analysis was performed by modeling 20 beam elements with an element length of 50 mm, and the numerically calculated data were used as measurements. An impact hammer was applied at a position 500 mm from the left end, which is the mid-span. The measurements were only taken at the mid-span and at two positions of 200 mm intervals from the left end.

Figure 4 shows the FRF receptance magnitude at the mid-span using a linear scale on the *x*-axis and a base-10 logarithmic scale on the *y*-axis when the frequency was incremented by 0.02 Hz. Three resonance frequencies were clearly observed in the plot, because a noise-free state was considered. In this figure, the lowest three resonance frequencies were 11.62, 34.72, and 58.5 Hz, respectively, and the data within a narrow frequency range at each resonance frequency were extracted to estimate the dynamic parameters. The parameters at positions 1, 2, and 3 at 200, 400, and 500 mm, respectively, from the left end were predicted using Equation (8a,b), respectively, as follows:$$\begin{array}{l} k^{31}=2.46\times 10^{5}\;N/\mathrm{mm},\;k^{32}=1.37\times 10^{5}\;N/\mathrm{mm},\;k^{33}=1.66\times 10^{5}\;N/\mathrm{mm},\;\\ M^{31}=8.79\;\mathrm{kg},\;M^{32}=5.38\;\mathrm{kg},\;M^{33}=4.96\;\mathrm{kg},\;\\ C^{31}=2.01\times 10^{2\;}N\cdot \sec/\mathrm{mm},\;C^{32}=2.80\times 10^{2}\;N\cdot \sec/\mathrm{mm},\\ C^{33}=4.01\times 10^{2}\;N\cdot \sec/\mathrm{mm}.\; \end{array}$$

These numerical values represent the parameter values at the measurement positions due to the impact at position 3. Repeating the measurement process due to the impact at the other locations, the parameter matrices of the finite element model can be completed. The updated parameter matrices can be saved and utilized for the subsequent maintenance of structural members.

Table 1 represents the FRF data range selected in the neighborhood of the resonance frequencies. It exhibits the elastic modulus ratio at positions 1, 2, and 3 at 200, 400, and 500 mm, respectively, from the left end, calculated by Equation (10a,b), with respect to the actual elastic modulus. In the table, the columns corresponding to $\Omega_{1}$, $\Omega_{2}$, and $\Omega_{3}$ represent the elastic moduli calculated by the SRSS of Equation (9). Taking the FRFs only within the first resonance frequency, the elastic modulus ratio was highly underestimated. The predicted ratio was greatly increased by considering the first two resonance states. Taking into account the FRF data up to the three resonance frequencies at the mid-span, the elastic modulus was estimated to be close to 97.2% of the actual value. They were predicted to be 50.7% and 71.9% at positions 1 and 2, respectively. It was observed that the predicted parameters were underestimated with the increase in the distance to the measurement location from the impact position. It was analyzed that the physical property can be very close to the actual value in taking the FRFs at the impact position.

### 3.2. A Cantilevered Steel Beam

This example predicted the physical properties of a cantilevered beam under the same conditions as the previous example (Figure 5). The measurements were conducted at positions 1, 2, 3, 4, and 5 at 200, 400, 600, 800, and 1000 mm from the fixed end, respectively. The impact was given at the position of 1000 mm, which was the free end. Figure 6 exhibits the FRF receptance magnitude curve according to the excitation frequency. Five resonance frequencies were clearly displayed in the plot, because noise-free FRF data were utilized. Compared with the previous both-end-fixed beam, the resonance frequencies of the cantilevered beam were distributed across a wider range. It is shown that the cantilevered beam was more flexible than the both-end-fixed beam. This example utilized the FRFs in the first four resonance states and predicted the following parameters using Equation (8a,b):$$\begin{array}{l} k^{51}=1.07\times 10^{4\;}N/\mathrm{mm},\;k^{52}=0.57\times 10^{4\;}N/\mathrm{mm},\;k^{53}=0.41\times 10^{4}\;N/\mathrm{mm},\;k^{54}=0.38\times 10^{4}\;N/\mathrm{mm},\;k^{55}=0.29\times 10^{4}\;N/\mathrm{mm},\\ M^{51}=16.4\;\mathrm{kg},\;M^{52}=8.66\;\mathrm{kg},\;M^{53}=6.18\;\mathrm{kg},\;M^{54}=5.72\;\mathrm{kg},\;M^{55}=4.40\;\mathrm{kg},\\ C^{51}=0.019\times 10^{3}\;N\cdot \sec/\mathrm{mm},\;C^{52}=0.010\times 10^{3}\;N\cdot \sec/\mathrm{mm},\\ C^{53}=4.12\times 10^{3\;}N\cdot \sec/\mathrm{mm},\;C^{54}=3.83\times 10^{3}\;N\cdot \sec/\mathrm{mm},\\ C^{55}=2.93\times 10^{3\;}N\cdot \sec/\mathrm{mm}. \end{array}$$

Table 2 displays the FRFs in the selected frequencies. It shows that the elastic modulus ratio taking the FRFs up to the lowest four resonance states at the impact position was in the range of 106.2~109.9% of the actual value. This indicates that the physical property can be close to the actual value, even if the FRF data are taken only in the first resonance state. Moreover, it was observed that the physical property at position 4 next to the impact position was also close to the actual value, despite the utilization of the FRFs only in the first resonance frequency. It was observed that the more flexible the structural member, the closer the physical property that can be obtained, despite taking the FRF data at the impact position in the first resonance state.

### 3.3. Estimation of Concrete Strength on an RC Column

The validity of the proposed method was investigated in predicting the compressive strength of concrete in an RC structure by a nondestructive test. The measured FRFs were contaminated by external noise. An impact hammer test was conducted to estimate the elastic modulus and compressive strength of concrete on an RC column with a cross-section of 400×400 mm and a length of 3.2 m (Figure 7). The compressive strength and elastic modulus of the concrete were assumed from the concrete core tests as 27 MPa and 24 GPa, respectively. Three accelerometers were installed at the mid-height of the column and at intervals of 300 mm down from the mid-height of the column. The FRF data at the mid-height of the column to correspond to the impact position were utilized to estimate the concrete’s strength. The column was assumed as a both-end-fixed column, and its stiffness could be calculated by Equation (10a).

The tests were repeated four times, and the FRF data were collected and saved for each test. Figure 8 presents the magnitude curves of the FRF inertance in a base-10 logarithmic scale according to an excitation frequency in the 0~500 Hz range. It was found that the four test curves of T-1, T-2, T-3, and T-4 had almost similar forms, and the data were greatly contaminated by noise due to the measurement and the inhomogeneous concrete. The contaminated FRF data will lead to inaccurate concrete strength results. The noise should be minimized by using denoised FRFs or averaging the FRF data from repeated experiments. This study reduced the noise effect by extracting the FRF data within a wide range of disturbance frequencies and averaging the corresponding results.

The lowest two resonance frequencies, as shown in Figure 8, were found to be 88 and 328 Hz, respectively. In this experiment, the FRF data in the 48.706~99.438 Hz range were extracted, including the first and second resonance frequencies. The FRFs after the second resonance frequency were disregarded in estimating the physical parameters, because the noise effect was greatly increased. Using Equations (10a) and (11) with the extracted FRFs, respectively, the predicted elastic modulus and the compressive strength of the concrete were calculated and are given in Table 3. The T-2 and T-4 test results were very close to the concrete properties, and the T-1 and T-3 results exhibited large differences. The main cause of the difference was the existence of external noise. Despite the variance, the averaged elastic modulus and compressive strength of the concrete were close to the values predicted during the concrete core tests. The values for the compressive strength of the concrete indicated the average strength in a certain region, because concrete is a nonhomogeneous material. It was found from this experiment that close concrete properties can be obtained by taking the FRFs in a wide frequency range and averaging the parameters through repeated tests. It is expected that more reasonable research results to validate the proposed method will be derived with practice.

## 4. Conclusions

This work utilized FRF data in the frequency domain rather than modal characteristics. This study presented a method for estimating the physical properties of structural members with the impact hammer test. It was shown that the proposed algorithm could estimate the dynamic parameters as well as the physical properties without considering external noise. It provided an explicit mathematical form using the generalized inverse. It was observed in the numerical experiments estimating the elastic modulus on a steel beam that the predicted values were underestimated with the increase in the distance to the measurement location from the impact position. Moreover, the physical properties were very close to the actual value as measured at the impact position. In the experiment to predict the compressive strength of the concrete on an RC column, it was predicted using the FRFs within a wide range of excitation frequencies including the first and second resonance frequencies. The predictions were challenging due to the existence of external noise contained in the measurement data. Close concrete properties can be obtained by analyzing the FRFs in a wide frequency range, by repeating the tests, and by averaging the test results. Theoretical analysis does not take into account the noise effect, but in practice, the external noise will lead to inconsistent results. It is expected that more comprehensive research will be performed in order to apply this method in practice and overcome the limitations of this study.

## Figures and Tables

**Figure 1 sensors-22-06762-f001:**
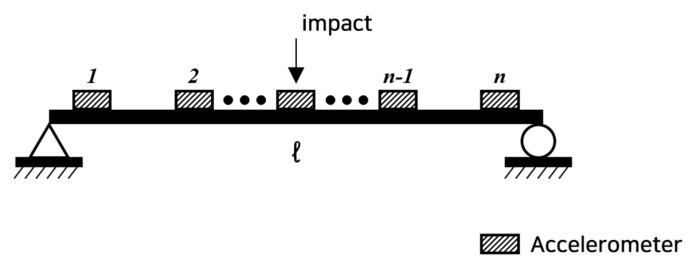
Layout of the impact hammer and accelerometers.

**Figure 2 sensors-22-06762-f002:**
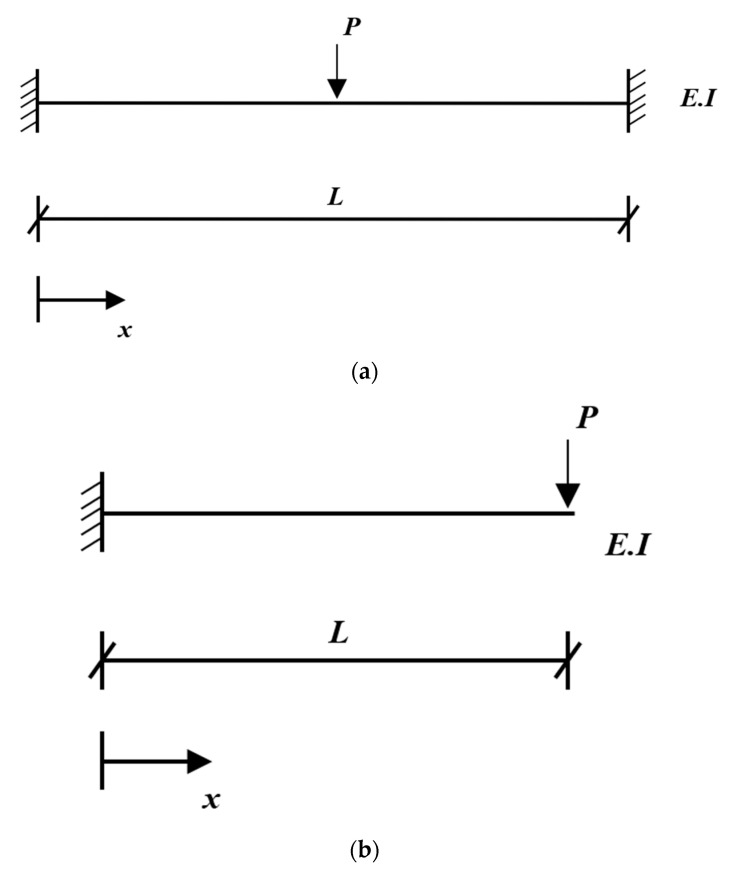
Elastic deflection curve: (**a**) both-end-fixed beam; (**b**) cantilevered beam.

**Figure 3 sensors-22-06762-f003:**
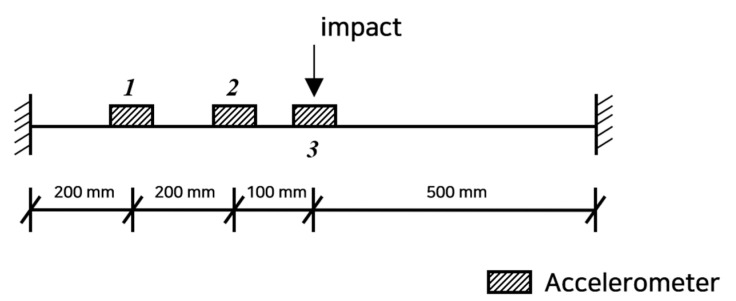
Steel beam of a finite element model with accelerometers and impact hammer.

**Figure 4 sensors-22-06762-f004:**
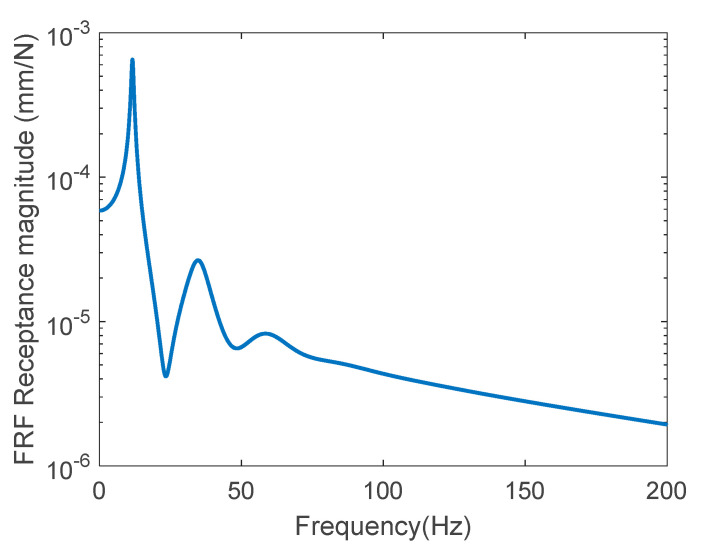
FRF receptance magnitude curve of a noise-free state.

**Figure 5 sensors-22-06762-f005:**
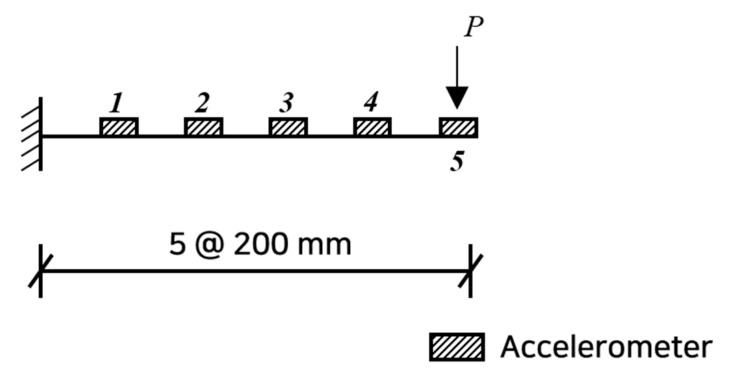
A cantilevered beam subjected to an impulse force at the free end.

**Figure 6 sensors-22-06762-f006:**
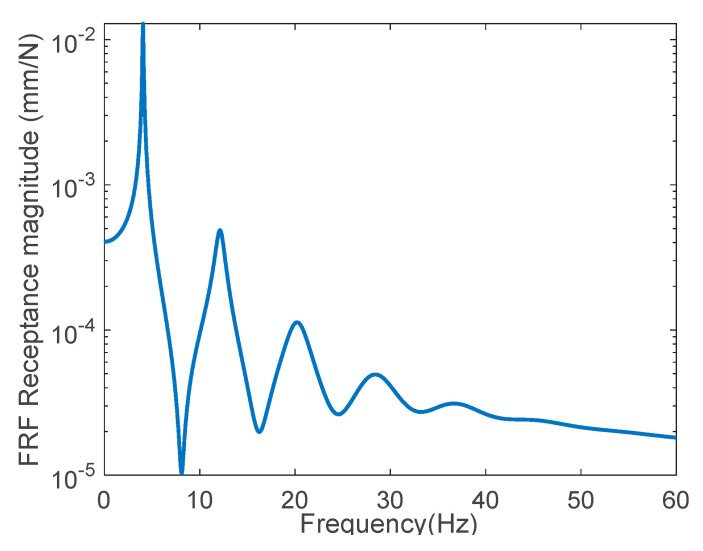
FRF receptance curve of a cantilevered beam.

**Figure 7 sensors-22-06762-f007:**
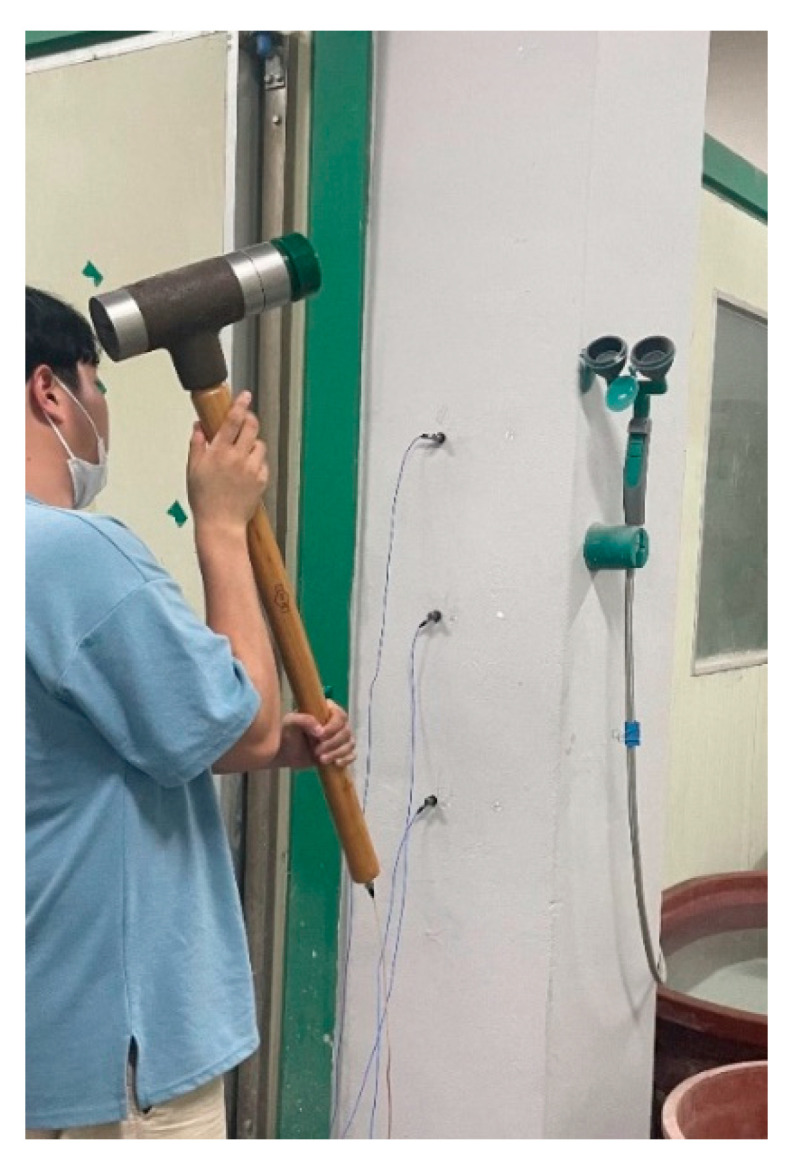
Impact hammer test.

**Figure 8 sensors-22-06762-f008:**
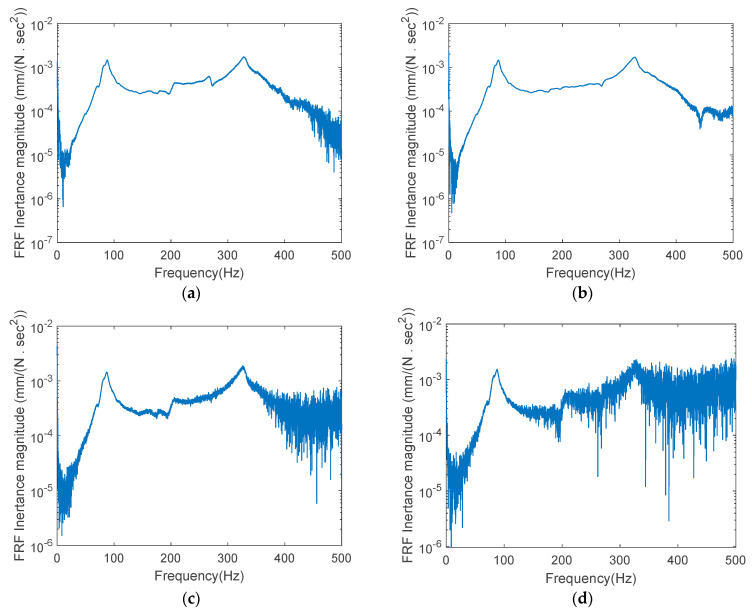
FRF inertance curve by the impact hammer test: (**a**) T-1; (**b**) T-2; (**c**) T-3; (**d**) T-4.

**Table 1 sensors-22-06762-t001:** Summary of the impact hammer test on a both-end-fixed beam.

**Distance (** * **x** * **)**	$\Omega_{1}$ **(11.62 Hz)**	$\Omega_{2}\;$ **(34.72 Hz)**	$\Omega_{3}\;$ **(58.5 Hz)**
Range of FRFs	11.58~11.62 Hz	34.68~34.74 Hz	53.98~56.98 Hz
200	0.0741	0.4060	0.5070
400	0.1158	0.6968	0.7192
500	0.1221	0.8782	0.9722

**Table 2 sensors-22-06762-t002:** Summary of the impact hammer test on a cantilevered beam.

**Distance (** * **x** * **)**	$\Omega_{1}$ **(4.04 Hz)**	$\Omega_{2}\;$ **(12.12 Hz)**	$\Omega_{3}\;$ **(20.22 Hz)**	$\Omega_{4}\;$ **(28.42 Hz)**
Range of FRFs	3.18~3.98 Hz	12.08~12.12 Hz	20.18~20.22 Hz	28.34~28.38 Hz
200	0.2217	0.2228	0.2236	0.2252
400	0.4265	0.4290	0.4432	0.4471
600	0.6267	0.6543	0.6603	0.6670
800	0.8343	0.8485	0.9798	1.0126
1000	1.0621	1.0707	1.0831	1.0991

**Table 3 sensors-22-06762-t003:** Estimated elastic modulus and compressive strength of the concrete.

**Distance (*x*)** **from the Bottom of the Column (mm)**	$\frac{E_{c}\;\left(GPa\right)}{f_{c}^{\prime }}\left(MPa\right)$				**Average** $E_{c}\;\left(GPa\right)/f_{c}^{\prime }\left(MPa\right)$
	**T-1**	**T-2**	**T-3**	**T-4**	
1600	14.6/9.7	24.7/27.8	32.1/46.6	23.8/25.6	23.8/27.4

## Data Availability

The data used to support the findings of this study are included within the article.

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
