# Peer review of "Prediction of the Physical Properties of a Structural Member by the Impact Hammer Test"

_sensors, 2022, doi:10.3390/s22186762_

Round 1
Reviewer 1 Report
This paper is fascinating research studying the mechanical properties of structural elements using the acquired dynamic response of the section. However, this study requires a major revision due to the fact that the introduction is missing some vital information and a literature review about structural system identification methods based on dynamic responses of the structure and structural health monitoring applications. Also, throughout the paper, there are many uncited paragraphs. Therefore, the paper results or references must back up every statement.
Abstract:
I am missing a sentence that clearly mentions the paper's novelty in the abstract. I suggest you to mention your contribution better. The way it is written is not very clear. Other authors have already used the dynamic response of a structure to estimate the mechanical characteristics of a structure. DOI: 10.1016/j.jsv.2020.115368
The paragraph between lines 26 and 33 requires citations. I suggest you write about the importance of structural system identification and structural health monitoring. I suggest you study the paper entitled “Low-Cost Wireless Structural Health Monitoring of Bridges”. This can help you enrich your literature review.
The paragraph between lines 39 and 46 requires citations. So please back your assumption up by adding some references. Also, I encourage you to spend a few words on the modal analysis of the structures using an impact hammer a little more.
The paragraph between lines 64 and 72 requires citations. Please back your assumption up by adding some references.
Before introducing your research, you should explain the reviewed methods' drawbacks and the gaps. Unfortunately, I am missing a paragraph clearly stating the novility of this paper and its advantages over available methods such as the dynamic observability method (DOI: 10.1016/j.jsv.2020.115368).
I am missing a paragraph at the end of your introduction with the paper organization.
Paragraph ( line 91 and 96) needs references.
Is figure 1 really necessary?
The equations are not following the template of the Sensor journal. Please review them.
The paper requires some editing, I see many general issues with typography and following the templates. The tables
Table 1 and 2 are not clear. Please revise their general look.
It is stated that the physical properties of the steel cantilever beam are predicted. I see no comparison of the predicted values with actual values. The only presented comparison is for the RC beam which has significant errors.
The current system is not very accurate. What would be the use of the developed method?
The estimated concrete strength varies between 9.7 and 46.6 MPa. The accuracy of the current system in its current form is not enough for SHM applications. How can the error be minimized?
Last recommendation: Do not use the personal pronoun (We) in scientific writing.
Reviewer 2 Report
The current paper presents an approach for estimating the physical characteristics of structural members through the impact hammer test. The reviewer comments are suggested as follows:
1- The main novelties of it are not clear to readers and reviewers. Authors are encouraged to add more comments on the novelties and main contributions of the present paper in the abstract section and last paragraph of the introduction section.
2- What are the applications of the proposed method? Authors are suggested to provide some technical expressions on applying the presented algorithm and needing this new finding.
3- The introduction section is very briefly organized. Authors are invited to revise the introduction section by reviewing some related works such as: doi.org/10.1016/j.compstruc.2021.106639. doi.org/10.1155/2022/9851533. doi.org/10.3390/pr10040656.
4- Authors are suggested to define all variables after their first appearance in the paper. Some more information on Eqs. (4-6) are needed. Authors are suggested to define all contributed terms and role of each term comprehensively.
5- What are the limitations, assumptions, and advantages of the proposed materials and model used in this paper?
6- The discussion is not informative and is very briefly organized. It should be enriched by adding more important conclusions and scientific interpretations.
Round 2
Reviewer 1 Report
My comments are well answered.
Reviewer 2 Report
All the requested modifications have been made. The article is acceptable as it is and can be published in its current form.